# Adult zebrafish Langerhans cells arise from hematopoietic stem/progenitor cells

Sicong He[1†], Jiahao Chen[2†], Yunyun Jiang[2], Yi Wu[3], Lu Zhu[3], Wan Jin[3], Changlong Zhao[3], Tao Yu[4], Tienan Wang[3], Shuting Wu[3], Xi Lin[3], Jianan Y Qu[1], Zilong Wen[3], Wenqing Zhang[2,5]*, Jin Xu[5]*

[1]Department of Electronic and Computer Engineering, Center of Systems Biology and Human Health, Hong Kong University of Science and Technology, Hong Kong, China; [2]Key Laboratory of Zebrafish Modeling and Drug Screening for Human Diseases of Guangdong Higher Education Institutes, Department of Developmental Biology, School of Basic Medical Sciences, Southern Medical University, Guangzhou, China; [3]Division of Life Science and State Key Laboratory of Molecular Neuroscience, Center of Systems Biology and Human Health, Hong Kong University of Science and Technology, Hong Kong, China; [4]Shenzhen Key Laboratory for Neuronal Structural Biology, Biomedical Research Institute, Shenzhen Peking University, The Hong Kong University of Science and Technology Medical Center, Shenzhen, China; [5]Division of Cell, Developmental and Integrative Biology, School of Medicine, South China University of Technology, Guangzhou, China

*For correspondence:
mczhangwq@scut.edu.cn (WZ);
xujin@scut.edu.cn (JX)

[†]These authors contributed equally to this work

Competing interests: The authors declare that no competing interests exist.

**Abstract** The origin of Langerhans cells (LCs), which are skin epidermis-resident macrophages, remains unclear. Current lineage tracing of LCs largely relies on the promoter-Cre-LoxP system, which often gives rise to contradictory conclusions with different promoters. Thus, reinvestigation with an improved tracing method is necessary. Here, using a laser-mediated temporal-spatial resolved cell labeling method, we demonstrated that most adult LCs originated from the ventral wall of the dorsal aorta (VDA), an equivalent to the mouse aorta, gonads, and mesonephros (AGM), where both hematopoietic stem cells (HSCs) and non-HSC progenitors are generated. Further fine-fate mapping analysis revealed that the appearance of LCs in adult zebrafish was correlated with the development of HSCs, but not T cell progenitors. Finally, we showed that the appearance of tissue-resident macrophages in the brain, liver, heart, and gut of adult zebrafish was also correlated with HSCs. Thus, the results of our study challenged the EMP-origin theory for LCs.
DOI: https://doi.org/10.7554/eLife.36131.001

## Introduction

Langerhans cells (LCs) are resident macrophages in skin epidermis, where they have immune-stimulatory capacities. Under both steady and inflammatory conditions, upon activation, LCs migrate from the epidermis to the skin-draining lymph nodes to present antigens to T cells, resulting in the activation of acquired immunity (*Hemmi et al., 2001*; *Merad et al., 2008*; *Romani et al., 2010*; *Stoitzner et al., 2005*). In addition, several studies have suggested that LCs also confer immune tolerance of the skin (*Bobr et al., 2010*; *Chopin and Nutt, 2015*; *Kaplan et al., 2005*; *Merad et al., 2008*; *Romani et al., 2012*; *Shklovskaya et al., 2011*; *van der Aar et al., 2013*). The biological significance of the opposing roles of LCs remains unclear, but it is thought that LCs dampen the immune response under steady-state conditions, but activate the response upon challenge (*Seneschal et al., 2012*).

Although LCs were identified over 100 years ago, their origins have been debated ever since their discovery. Initially, LCs were considered to be nerve cells in the skin (*Langerhans, 1868*) and subsequently were thought to have a melanocyte or keratinocyte origin (*Birbeck et al., 1961*; *Reams and Tompkins, 1973*). The hematopoietic origin of LCs was not revealed until 1979, when researchers observed that donor-derived LCs were present in the skin of lethally-irradiated mice transplanted with allogeneic or semi-allogeneic hematopoietic precursors (*Frelinger et al., 1979*; *Katz et al., 1979*). This led to the hypothesis that LCs were replenished continually by circulating monocytes (*Romani et al., 2012*). However, this view was challenged by several subsequent studies. The first clue came from transplantation studies by Merad et al., who reported that most LCs in lethally-irradiated mice reconstituted with either congenic hematopoietic precursors or T cell-free allogenic precursors were of a host origin (*Merad et al., 2004*; *Merad et al., 2002*). Subsequent studies further documented that LCs turn over slowly in steady-state. In fact, LCs are replenished by a small number of proliferative LC precursors residing in the skin, although circulating precursors do contribute to LCs when a severe injury occurs (*Chorro et al., 2009*; *Collin et al., 2006*; *Ghigo et al., 2013*; *Gomez Perdiguero et al., 2015*; *Hoeffel et al., 2015*; *Hoeffel et al., 2012*; *Kanitakis et al., 2004*; *Krueger et al., 1983*; *Schulz et al., 2012*; *Vishwanath et al., 2006*; *Yona et al., 2013*). Despite these studies, the origin and nature of the proliferative LC precursors remain largely undefined.

The self-renewal ability of adult LCs led to the hypothesis that LCs are generated from sources independent of hematopoietic stem cells (HSCs), which are born in the aorta, gonads, and mesonephros (AGM) in mammals (*Orkin and Zon, 2008*). This was supported by a recent fate-mapping study by Hoeffel et al., who utilized *Runx1-MER-Cre-MER/Rosa26-YFP* reporter mice and showed that adult LCs in mice had dual origins: YS primitive monocytes and fetal liver monocytes (*Hoeffel et al., 2012*). Further fate-mapping studies with similar *MER-Cre-MER/Rosa26* reporter systems suggested that adult LCs in mice were predominantly generated from YS-derived erythro-myeloid precursors (EMPs) (*Gomez Perdiguero et al., 2015*; *Hoeffel et al., 2015*). Yet, this EMP-origin theory was challenged by a recent study by Sheng et al., who utilized the *c-Kit-MER-Cre-MER/Rosa26-YFP* reporter system to trace the origin of tissue-resident macrophages and found that most resident macrophages, including LCs, in adult mice were predominantly derived from HSCs but not from EMPs (*Sheng et al., 2015*). However, despite their elegant designs, these fate-mapping studies, relied on promoter-controlled CreER-*loxP* tracking systems. The exact transcription activity of these promoters in the tissue of interest remains to be further elucidated, so such studies cannot provide a definitive answer about the origin of LCs. Furthermore, conventional lineage-tracing systems cannot selectively label and distinguish cells from different anatomic locations. These shortcomings have hindered the identification of the origin of LCs, so a new cell labeling strategy that can provide both temporal and spatial resolution is required.

Similar to mammals, zebrafish experience multiple waves of hematopoiesis (*Jagannathan-Bogdan and Zon, 2013*; *Jing and Zon, 2011*; *Stachura and Traver, 2011*; *Xu et al., 2012*). The first or embryonic hematopoiesis in the zebrafish initiates at ~11 hr post fertilization (hpf) in the posterior lateral mesoderm (PLM) and rostral blood island (RBI), which are, similar to the mammalian yolk sac (YS), producing embryonic erythroid and myeloid cells respectively. The second or definitive wave of hematopoiesis occurs at ~28 hpf in the ventral wall of the dorsal aorta (VDA), a tissue equivalent to the mammalian AGM (*Orkin and Zon, 2008*), and gives rise to HSCs capable of generating all blood cell types during fetal life and adulthood. A third or intermediate wave of hematopoiesis, which generates EMPs, is believed to initiate autonomously from the posterior blood island (PBI) at around 30 hpf and produces erythroid and myeloid cells during both embryonic and fetal development (*Bertrand et al., 2007*). Thus, its conserved hematopoietic program, genetic amenability, and imaging feasibility have made zebrafish an excellent model system to use for fate-mapping studies of LCs.

In the current study, we utilized the recently developed temporospatially resolved cell labeling IR-LEGO-CreER-*loxP* system (*Deguchi et al., 2009*; *Kamei et al., 2009*; *Xu et al., 2015*), together with genetic analysis, to delineate the origins of adult LCs in zebrafish. We showed that most LCs in adult zebrafish are derived from the VDA, a tissue equivalent to the mouse AGM, from where HSCs emerge. Genetic studies showed that VDA-derived LCs required runx1 and cMyb functions. Further fate-mapping analyses showed that adult LCs were largely accompanied by HSCs but not non-HSC T cell progenitors. Finally, we showed that other adult tissue-resident macrophages in the brain,

liver, heart, and gut are also correlated with HSCs, suggesting that primary adult tissue-resident macrophages were likely derived from HSCs in zebrafish.

## Results

### Adult LCs in zebrafish arise predominantly from the VDA region

The IR-LEGO system was previously demonstrated to provide high-resolution temporospatial cell labeling in zebrafish (*Deguchi et al., 2009*; *Kamei et al., 2009*; *Xu et al., 2015*). To apply this system to study the origin of LCs in zebrafish, we generated a *Tg(mpeg1:loxP-DsRedx-loxP-GFP)* transgenic line, in which macrophages, including LCs, were specifically marked by DsRedx under normal conditions but by GFP upon *loxP* excision (*Ellett et al., 2011*). The transgenic fish were outcrossed with a heat shock-inducible *Tg(hsp70:mCherry-T2a-CreER$^{T2}$)* line (*Hans et al., 2011*) and the resulting double transgenic fish *Tg(hsp70:mCherry-T2a-CreER$^{T2}$;mpeg1:loxP-DsRedx-loxP-GFP)* were used for LC fate mapping analysis (*Figure 1—figure supplement 1*). To track the contribution of all hematopoietic sources to LCs, we locally heat-shocked the RBI, VDA, and PBI, the hematopoietic tissues for the generation of embryonic macrophages, HSCs, and EMPs respectively (*Jagannathan-Bogdan and Zon, 2013*; *Jing and Zon, 2011*; *Stachura and Traver, 2011*; *Xu et al., 2012*), with a 1342–1350 nm infrared (IR) laser followed by treatment with 4-OH tamoxifen (4-OHT) (*Figure 1—figure supplement 1*). The RBI region was labeled at 14–16 hpf when embryonic myelopoiesis begins (*Ciau-Uitz et al., 2014*), the VDA region was labeled at 28 hpf when the endothelial hematopoietic transition (EHT) occurs (*Kissa and Herbomel, 2010*), and the PBI region was heat-shocked at 20 hpf to prevent contamination of the RBI- and VDA-derived cells by the circulation which begins at ~25 hpf (*Figure 1—figure supplement 1C and D*) (*Ciau-Uitz et al., 2014*). The heat-shocked embryos were raised to 4 days post fertilization (dpf) and adulthood, and the contribution from each labeled region to LCs was determined by anti-GFP staining. To identify LCs, we co-stained mpeg1-transgenic DsRedx with krtt1c19e-transgenic GFP in *mpeg1* and *krtt1c19e* double transgenic fish, which primarily marks the basal epidermal layer (*Lee et al., 2014*). We found that most DsRedx+ macroophages on the superficial part of the adult skin located within the krtt1c19e+ epidermal layer, suggesting that they were indeed LCs (*Figure 1—figure supplement 1E*). We then considered the mpeg1-transgenic DsRedx/GFP cells on the superficial part of the skin as LCs in the rest of the study. Tracing results showed that, although the RBI was the primary source for LCs at early embryonic stages, most (~80% LCs are GFP$^+$) LCs in the adult zebrafish (≥3 months old) were of a VDA origin and only limited numbers of adult LCs were derived from the RBI and PBI, compared with non-heat-shocked control fish (*Figure 1B–D*, *Figure 1—source data 1*). Although we attempted to label the whole VDA region, we were unable to guarantee that every hematopoietic stem/progenitor cell (HSPC) was successfully labeled. According to the tracing result that ~80% of adult LCs were labeled, we hypothesized that our labeling efficiency was also about 80%. The remaining ~20% non-GFP +LCs were likely derived from unlabeled HSPCs in VDA due to the incomplete heat-shock or *loxP* recombination. To check when LCs shifted from a RBI origin to a VDA origin, heat-shocked embryos were examined from 4 dpf to 12 dpf at 2 day intervals (*Figure 1—figure supplement 1F*). The results showed that the percentage of RBI-derived LCs gradually decreased and only ~7% of LCs were derived from a RBI origin at 12 dpf (*Figure 1—figure supplement 1G and H*, *Figure 1—figure supplement 1—source data 1*). By contrast, the VDA-derived LCs became obvious at 6 dpf and their percentage reached 54% by 12 dpf (*Figure 1—figure supplement 1G and H*, *Figure 1—figure supplement 1—source data 1*). A small portion of PBI-derived LCs was observed from 6 dpf. In contrast to the prosperous LCs from the VDA, the PBI-derived LCs maintain their percentage at ~15% from 8 dpf to 12 dpf suggesting limited cell proliferation or input from the PBI (*Figure 1—figure supplement 1G and H*, *Figure 1—figure supplement 1—source data 1*). Taken together, these data suggested that LCs gradually shifted from a RBI origin to a VDA origin during the early larval stages of zebrafish development. Eventually, most adult LCs were derived from the VDA region in the zebrafish.

### VDA-derived LCs require Runx1 and cMyb functions

Given that the zebrafish VDA region is equivalent to the mouse AGM where HSCs are born (*Orkin and Zon, 2008*), we reasoned that a large portion of adult LCs in the zebrafish were likely to

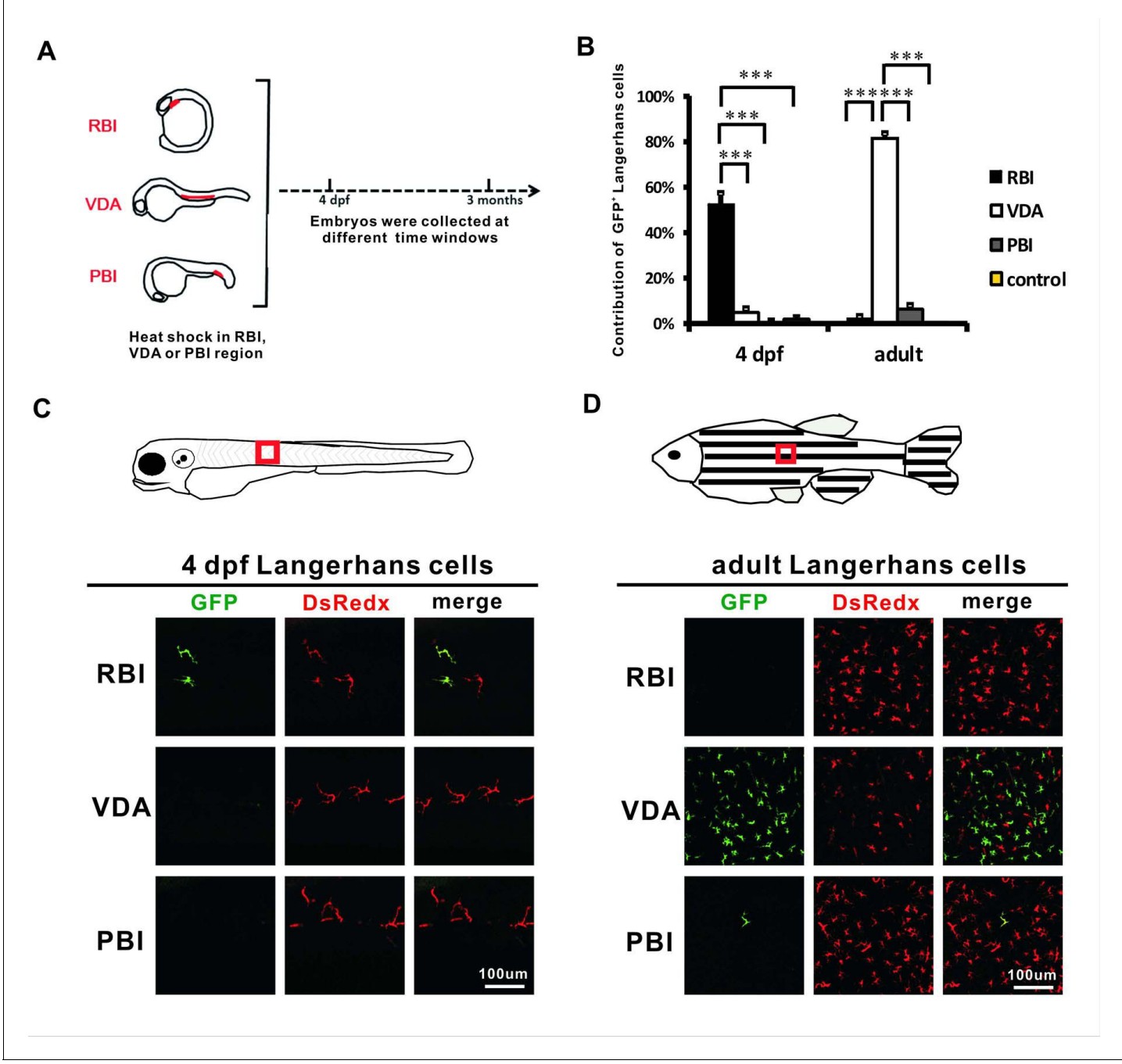

**Figure 1.** Adult LCs are largely derived from the VDA region in zebrafish. (**A**) A schematic diagram indicates the RBI, VDA and PBI regions and the time point (4 dpf and 3 months) when the heat-shocked fish are analyzed. (**B**) Quantification of the percentage of GFP+ LCs derived from the RBI, VDA, PBI, and non-heat-shocked control at 4 dpf and adulthood. n = 13, 7, 10 and 6 for the RBI-, VDA-, PBI-, and non-heat-shocked control fish analyzed at 4 dpf respectively. n = 5, 5, 5 and 6 for the RBI-, VDA-, PBI-, and non-heat-shocked control fish analyzed at adulthood, respectively. Error bars represent mean SEM. ***p<0.001. (**C**) Anti-GFP staining shows that GFP+ LCs are detected in the RBI-labelled fish, but not VDA- and PBI-labelled fish at 4 dpf. The red box indicates the imaging region. (**D**) Anti-GFP staining indicates that GFP+ LCs are mainly detected in the VDA-labelled fish, but not the RBI- and PBI-labelled fish in adulthood. The red box indicates the imaging region.

DOI: https://doi.org/10.7554/eLife.36131.002

The following source data and figure supplements are available for figure 1:

**Source data 1.** Quantification of GFP+ Langerhans cells at embryonic and adult stages.

DOI: https://doi.org/10.7554/eLife.36131.005

**Figure supplement 1.** Experimental design of the IR-LEGO-CreER-*loxP* cell labelling system for LC fate mapping.

*Figure 1 continued on next page*

*Figure 1 continued*

DOI: https://doi.org/10.7554/eLife.36131.003

**Figure supplement 1—source data 1.** Quantification of GFP$^+$ Langerhans cells at early developmental stages.

DOI: https://doi.org/10.7554/eLife.36131.004

be the progeny of HSPCs in the VDA. To test this hypothesis, we crossed *Tg(hsp70:mCherry-T2a-CreER$^{T2}$;mpeg1:loxP-DsRedx-loxP-GFP)* with *runx1$^{W84X}$* mutants, in which HSPC formation in the VDA region was severely impaired (*Bertrand et al., 2010*; *Boisset et al., 2010*; *Chen et al., 2009*; *Jin et al., 2009*; *Kissa and Herbomel, 2010*; *Sood et al., 2010*), and examined the VDA-derived LCs at ~12 dpf. Our results showed a reduction of >85% (7.0 cells/mm$^2$ in the mutants versus 53.8 cells/mm$^2$ in siblings) in the density of GFP$^+$ LCs in *runx1$^{W84X}$* mutants (*Figure 2A and B*, *Figure 2—source data 1*). The total density of LCs in *runx1$^{W84X}$* mutants was also reduced (57.7 cells/mm$^2$ in the mutants versus 173.2 cells/mm$^2$ in siblings) (*Figure 2—figure supplement 1E*, *Figure 2—figure supplement 1—source data 1*). By contrast, the RBI-derived GFP$^+$ LCs showed no obvious differences between mutants and siblings when heat-shock labeling was performed in the RBI region (*Figure 2—figure supplement 1C*, *Figure 2—figure supplement 1—source data 1*). This result was not surprising given that the RBI-derived macrophages are not reduced in *runx1$^{W84X}$* mutants (*Jin et al., 2012*).

To further test whether VDA-derived LCs were derived from HSPCs, we also examined these cells in *cmyb$^{hkz3}$* mutants with a *Tg(hsp70:mCherry-T2a-CreER$^{T2}$;mpeg1:loxP-DsRedx-loxP-GFP)* background, in which HSPCs were gradually depleted as a result of the inactivation of cMyb, a transcription factor essential for HSC development in mice and fish (*Mucenski et al., 1991*; *Mukouyama et al., 1999*; *Soza-Ried et al., 2010*; *Sumner et al., 2000*; *Zhang et al., 2011*). Given that *cmyb$^{hkz3}$* mutants have a developmental delay and cannot survive beyond 1 month of age (*Xu et al., 2015*), we assessed the LCs in 20 dpf *cmyb$^{hkz3}$* mutants, comparing them with size-matched 12–15 dpf siblings (*Figure 2—figure supplement 1B*, *Figure 2—figure supplement 1—source data 1*). Our results showed a partial reduction (15.2 cells/mm$^2$ in the mutants versus 40.4 cells/mm$^2$ in siblings) in the density of GFP$^+$ LCs in *cmyb$^{hkz3}$* mutants (*Figure 2C and D*, *Figure 2—source data 1*). The remaining GFP$^+$ LCs in the mutants might reflect either a cMyb-independent non-stem cell-derived population in VDA or differentiated progenies from the residual surviving HSCs. The total density of LCs in *cmyb$^{hkz3}$* mutants was also reduced (66.3 cells/mm$^2$ in the mutants versus 133.3 cells/mm$^2$ in siblings) (*Figure 2—figure supplement 1F*, *Figure 2—figure supplement 1—source data 1*). Unsurprisingly, there was no obvious difference in RBI-derived GFP$^+$ LCs between the mutants and siblings when heat-shock labeling was performed in the RBI region (*Figure 2—figure supplement 1D*, *Figure 2—figure supplement 1—source data 1*). This result was consistent with a previous report that cMyb was not required for the development of RBI-derived macrophages (*Jin et al., 2016*). Nevertheless, this result clearly showed that, although it was not absolutely necessary, cMyb was partially required for these cells. These data, together with the fate-mapping results, indicated that adult LCs in the zebrafish likely arose from HSPCs.

## The origin of adult LCs correlates spatially with HSCs

We previously showed that the VDA region contained non-HSC derived T cell progenitors (*Tian et al., 2017*). This finding prompted us to speculate that both HSCs and non-HSC progenitors in VDA give rise to adult LCs. To test which is the primary origin of adult LCs, we randomly labeled a single spot of the VDA region in *Tg(mpeg1:loxP-DsRedx-loxP-GFP);Tg(hsp70:mCherry-T2a-CreER$^{T2}$)* embryos (*Figure 3A*), with the hope that we might be able to separate non-HSC progenitors from HSCs. Labeled fish (a total of 13 fish) were raised to adulthood and the distribution patterns of VDA-derived LCs were determined by GFP expression. We found that 8 out of the 13 fish contained abundant GFP$^+$ LCs (*Figure 3B–C*, *Figure 3—source data 1*). Notably, abundant GFP$^+$ cells were also present in the circulation of these fish suggesting the successful labeling of HSCs (*Figure 3D and E*, *Figure 3—source data 1*). This group of fish was designated the 'HSC Group' (*Figure 3B–E*). In the remaining five VDA-labeled fish, GFP$^+$ LCs failed to be abundantly detected (*Figure 3B and C*) and few GFP$^+$ cells were found in the circulation, suggesting a failure of HSC labeling (*Figure 3D*

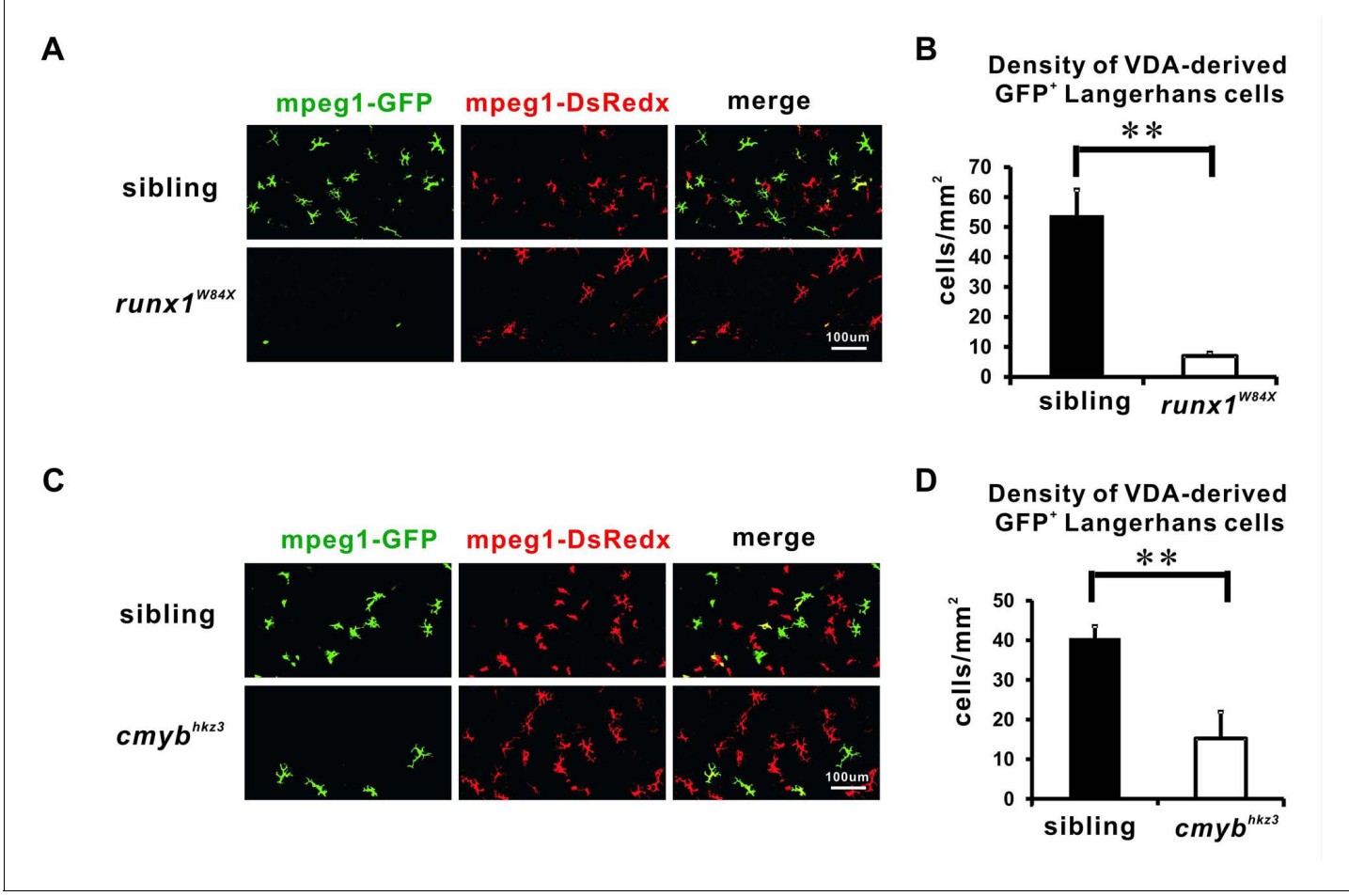

**Figure 2.** The VDA-derived LCs are reduced in Runx1 and cMyb mutants. (**A**) Anti-GFP staining shows that the VDA-derived LCs are significantly reduced in 12 dpf *runx1^{W84X}* mutants comparing with those in siblings. (**B**) Quantification of the density of the VDA-derived GFP$^+$ LCs in 12 dpf *runx1^{W84X}* mutants and siblings. n = 5 for siblings and n = 6 for mutants. Error bars represent mean SEM. **p<0.01. (**C**) Anti-GFP staining reveals a reduction of the VDA-derived LCs in 20 dpf *cmyb^{hkz3}* mutants comparing with size matching siblings. (**D**) Quantification of the density of the VDA-derived GFP$^+$ LCs in 20 dpf *cmyb^{hkz3}* mutants and size matching siblings. n = 11 and 10 for siblings and mutants respectively. Error bars represent mean SEM. **p<0.01.

DOI: https://doi.org/10.7554/eLife.36131.006

The following source data and figure supplements are available for figure 2:

**Source data 1.** Quantification of VDA derived LCs in Runx1 and cMyb mutants.
DOI: https://doi.org/10.7554/eLife.36131.009
**Figure supplement 1.** Quantification of body length and RBI derived LCs in Runx1 and cMyb mutants.
DOI: https://doi.org/10.7554/eLife.36131.007
**Figure supplement 1—source data 1.** Quantification of body length and RBI derived LCs in Runx1 and cMyb mutants.
DOI: https://doi.org/10.7554/eLife.36131.008

*and E*). These fish were defined as the 'non-HSC Group' (*Figure 3B–E*). These data indicated that the primary population of GFP$^+$ LCs observed in the VDA-labeled fish was associated with HSCs.

We previously showed that the density of HSCs gradually decreased from the anterior to the posterior VDA (*Tian et al., 2017*). If adult LCs arose from HSCs, they should follow this distribution pattern. To test this hypothesis, we artificially divided the VDA region into four positions, namely position 1 (P1) to position 4 (P4), from the anterior to posterior (*Figure 3F*) and heat-shocked only one position in each *Tg(mpeg1:loxP-DsRedx-loxP-GFP);Tg(hsp70:mCherry-T2a-CreER^{T2})* embryo. The heat-shocked embryos were then raised to adulthood, and the percentage of the adult fish

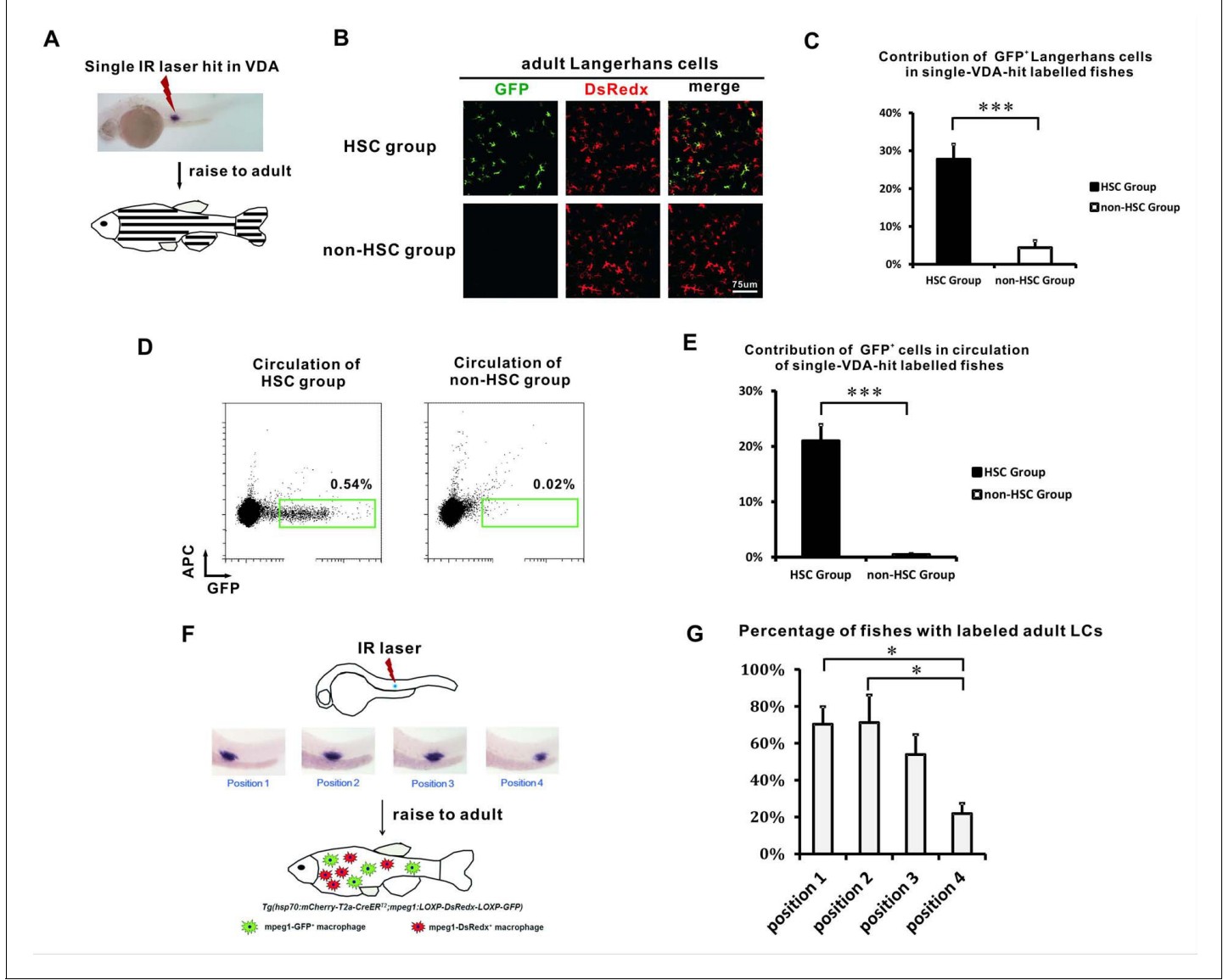

**Figure 3.** The VDA-derived LCs correlate with HSCs spatially. (**A**) A schematic view of the single spot VDA heat-shocked experiment. A single IR laser hit is performed in the VDA region. The heat-shocked embryos are raised to adult (about 3 months) for analysis. (**B**) GFP+ LCs are mainly found in the HSC group but not in the non-HSC group of VDA labeling. (**C**) Quantification of the relative contribution of GFP+ LCs in HSC (n = 8) and non-HSC (n = 5) groups. Error bars represent mean SEM. ***p<0.001. (**D**) Flow cytometry shows GFP+ cells are mainly found in the circulation of HSC group but not in the circulation of non-HSC group. (**E**) Quantification of the relative contribution of GFP+ cells in the circulation of HSC (n = 8) and non-HSC (n = 5) groups. Error bars represent mean SEM. ***p<0.001. (**F**) A schematic view of the position restricted single spot VDA heat-shocked experiment. The VDA region from anterior to posterior is artificially divided into four positions (P1 to P4) and each position is labeled by a single IR laser hit. The embryos are raised to adult and the percentage of successful LCs labelling is analyzed for each labeled position. (**G**) The percentage of fish with widely labeled LCs in each position. Three independent experiments were performed. There are total 35 fish for position 1, 44 fish for position 2, 40 fish for position 3, and 38 fish for position 4. *p<0.05.

DOI: https://doi.org/10.7554/eLife.36131.010

The following source data is available for figure 3:

**Source data 1.** Quantification data for *Figure 3C, E, and G*.

DOI: https://doi.org/10.7554/eLife.36131.011

containing GFP$^+$ LCs was calculated for each group. Our results showed that 70%, 71%, and 54% of the labeled fish contained abundant LCs for the P1, P2, and P3 groups, respectively. By contrast, only 22% of the labeled fish contained abundant LCs in the P4 group (*Figure 3G*, *Figure 3—source data 1*). This suggested that the generation of adult LCs was correlated with the density gradient of HSCs along the VDA.

## The generation of adult LCs correlates with HSCs but not non-HSC T cell progenitors

In addition to HSCs, the VDA of zebrafish also produces non-HSC progenitors that give rise to T cells (*Tian et al., 2017*). Adult LCs might share common progenitors with non-HSC-derived T cells. Non-HSC-derived T cells were reported to generate the first wave of T cells, whereas HSCs generate the second wave of T cells after 5 dpf (*Tian et al., 2017*). This phenomenon provides a temporal criterion to separate HSCs from non-HSC T cell progenitors. To do so, we utilized the *Tg(hsp70:mCherry-T2a-CreER$^{T2}$;coro1a:loxP-DsRedx-loxP-GFP)* fish in which the *coro1a* promoter drove the expression of GFP/DsRedx reporter genes in all leukocytes, including T cells and LCs (*Li et al., 2012*). The embryos of this double transgenic line were labeled at the P2 position at about ~24 hpf. All labeled embryos were raised to 5 dpf and the GFP contribution in thymocytes (visualized by coro1a:GFP in the thymus) was examined under a confocal microscope. The embryos were separated into two groups according to the GFP contribution to thymocytes. Group I contained abundant GFP$^+$ thymocytes (>10 cells). By contrast, Group II contained few GFP$^+$ cells (0–5 cells) suggesting a failure of labeling the first wave of T cells (*Figure 4A and B*). Embryos containing 5–10 GFP$^+$ thymocytes were discarded. Group I and II were raised to adulthood and their peripheral blood was analyzed. According to the GFP contribution to their peripheral blood, the fish were divided into two subgroups for each group. Group I-A and Group II-A contained significant GFP$^+$ cells in their peripheral blood, suggesting the successful labeling of HSCs (*Figure 4A,C and D*, *Figure 4—source data 1*). Group I-B and Group II-B contained few GFP$^+$ cells in their peripheral blood, suggesting that Group I-B only labeled non-HSC progenitors which could give rise to the first wave of T cells but not HSCs, and Group II-B represented a failure to label cells (*Figure 4A,C and D*, *Figure 4—source data 1*). Interestingly, only in Group I-A and Group II-A, but not in Group I-B and Group II-B, were significant GFP$^+$ cells found in the skin (*Figure 4E and F*, *Figure 4—source data 1*). Although we do not have LC antibodies to specifically distinguish LCs from other GFP$^+$ leukocytes, a significant portion of GFP$^+$ cells showed a ramified LC-like morphology (*Figure 4F*), suggesting that LCs were labeled in Group I-A and Group II-A. We then estimated the labeling efficiency by counting the GFP contribution in all leukocytes labeled by the anti-Lcp1 antibody (*Figure 4F*). We found 18–19% GFP$^+$ leukocytes in Group I-A and Group II-A compared with almost no GFP$^+$ leukocytes in Group I-B and Group II-B and the non-heat-shocked control groups (*Figure 4E and F*, *Figure 4—source data 1*). Taken together, our results suggested that adult LCs associated with HSCs but not non-HSC T cell progenitors.

## Adult tissue-resident macrophages in the liver, heart, gut, and brain also correlate with HSCs in lineage tracing

The HSC-associated origin of adult LCs prompted us to trace the origins of other tissue-resident macrophages in zebrafish. We also investigated whether other tissue-resident macrophages derived from RBI, PBI, or VDA. Similar to LCs and previously reported microglia (*Xu et al., 2015*), the primary adult tissue-resident macrophages in liver, gut, and heart were also derived from VDA, but not from RBI or PBI (*Figure 5—figure supplement 1A–D*, *Figure 5—figure supplement 1—source data 1*). To further differentiate whether these resident macrophages were generated by HSCs or not, we examined the contribution of GFP$^+$ macrophages in single-point VDA-labeled HSC and non-HSC groups (*Figure 3*). The results showed that GFP$^+$ macrophages in the liver, gut, heart and even brain were primarily found in the HSC group but not the non-HSC group (*Figure 5—figure supplement 1E*, *Figure 5—figure supplement 1—source data 1*), suggesting that tissue-resident macrophages in zebrafish spatially correlated with HSCs. To further test whether tissue-resident macrophages were derived from an HSC origin, Group I-A, I-B, II-A and II-B *Tg(hsp70:mCherry-T2a-CreER$^{T2}$;coro1a:loxP-DsRedx-loxP-GFP)* fish were assayed. Similar to the results of LCs, only fish labeled with HSCs (Group I-A and Group II-A) contained significant number of GFP$^+$ cells in brain,

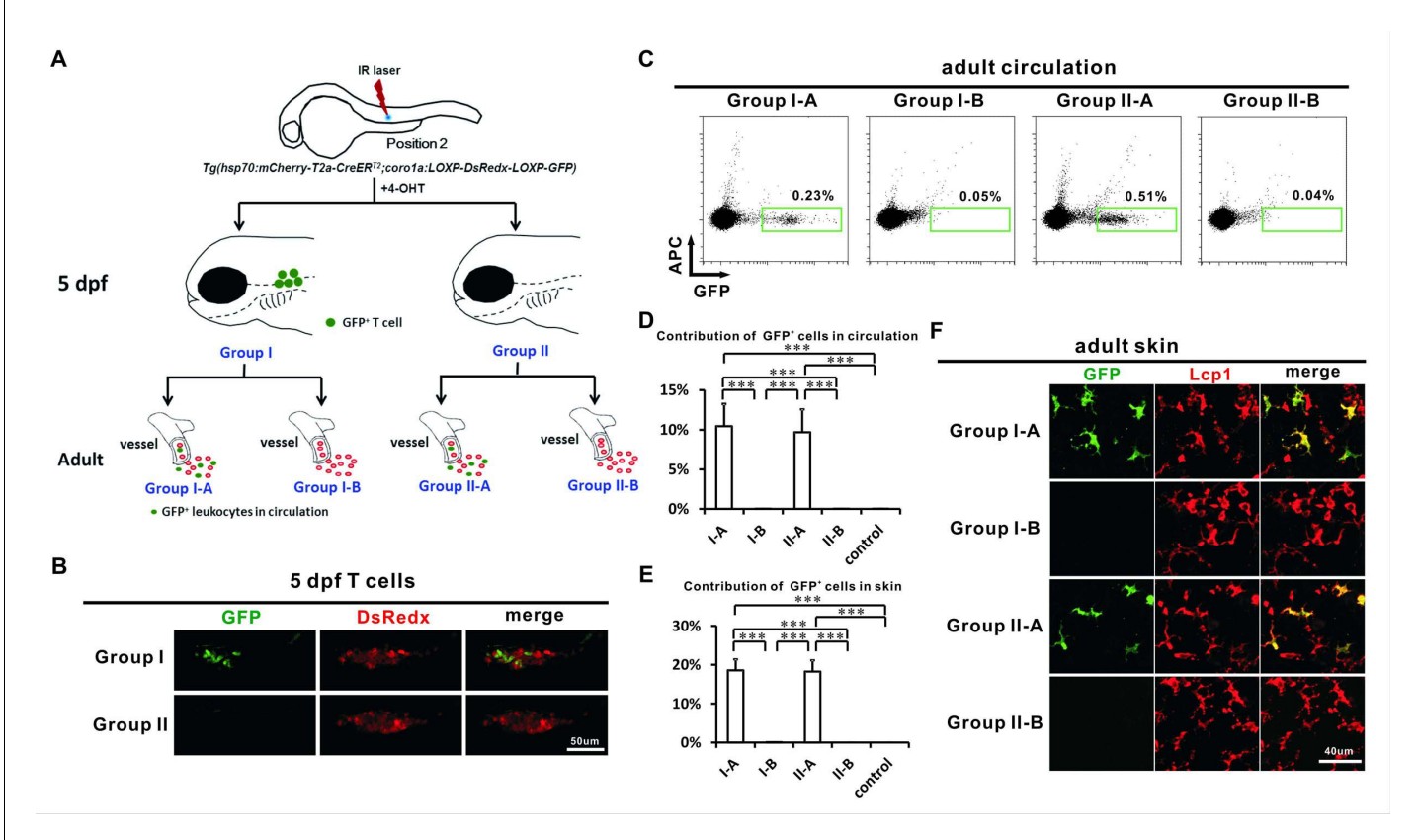

**Figure 4.** Adult LCs correlate with HSCs but not non-HSC progenitors. (**A**) A schematic view of the experimental design for laser labeling of *Tg(hsp70: mCherry-T2a-CreER^T2;coro1a:loxP-DsRedx-loxP-GFP)* embryos. A single IR laser hit was performed at Position 2 of VDA at about 24 hpf. After 4-OHT treatment, the heat-shocked embryos were raised to 5 dpf and the embryos were then separated into two groups according the GFP contribution to thymocytes. Group I embryos contain abundant GFP⁺ thymocytes, whereas Group II embryos hardly have GFP⁺ thymocytes. These two groups of embryos were raised to adult (over 3 months) and were subdivided into Group I-A, I-B, II-A and II-B according to the GFP contribution in circulation. (**B**) Confocal images show that Group I embryos, but not Group II embryos, contain abundant GFP⁺ thymocytes. (**C**) Flow cytometry shows that the circulation of Group I-A and II-A, but not I-B and II-B contain abundant GFP⁺ cells. (**D**) Quantification of the contribution of GFP⁺ cells in all fluorescent positive leukocytes in the circulation of Group I-A, I-B, II-A, II-B, and non-heat-shocked control group (n = 6 for each group). Error bars represent mean SEM. ***p<0.001. (**E**) Quantification of the contribution of GFP⁺ cells in all Lcp1 positive leukocytes in the skin of adult Group I-A, I-B, II-A, II-B, and non-heat-shocked control group (n = 5 for each group). Error bars represent mean SEM. ***p<0.001. (**F**) Confocal images show that GFP⁺ cells are mainly found on the skin of Group I-A and II-A, but not I-B and II-B. a significant portion of GFP⁺ cells showed ramified LC-like morphology, suggesting that they are LCs presumably.

DOI: https://doi.org/10.7554/eLife.36131.012

The following source data is available for figure 4:

**Source data 1.** Quantification data for *Figure 4D and E.*
DOI: https://doi.org/10.7554/eLife.36131.013

liver, heart, and gut (*Figure 5*, *Figure 5—source data 1*), suggesting that adult resident macrophages in brain, liver, heart, and gut were associated with HSCs in zebrafish. Interestingly, most GFP⁺ cells were Lcp1⁺, with the exception of gut GFP⁺ cells, which only partially overlapped with Lcp1 signals, suggesting that Lcp1 and coro1a marked different sub-populations of leukocytes in the gut. The identity of either Lcp1 or coro1a positive leukocytes warrants further study.

## Adult tissue-resident macrophages are largely associated with HSCs during transplantation

To further support the hypothesis that a substantial portion of adult tissue-resident macrophages were derived from HSCs, we performed cell transplantation experiments, in which hematopoietic cells isolated from the adult kidney of *Tg(mpeg1:loxP-DsRedx-loxP-GFP)* transgenic fish were

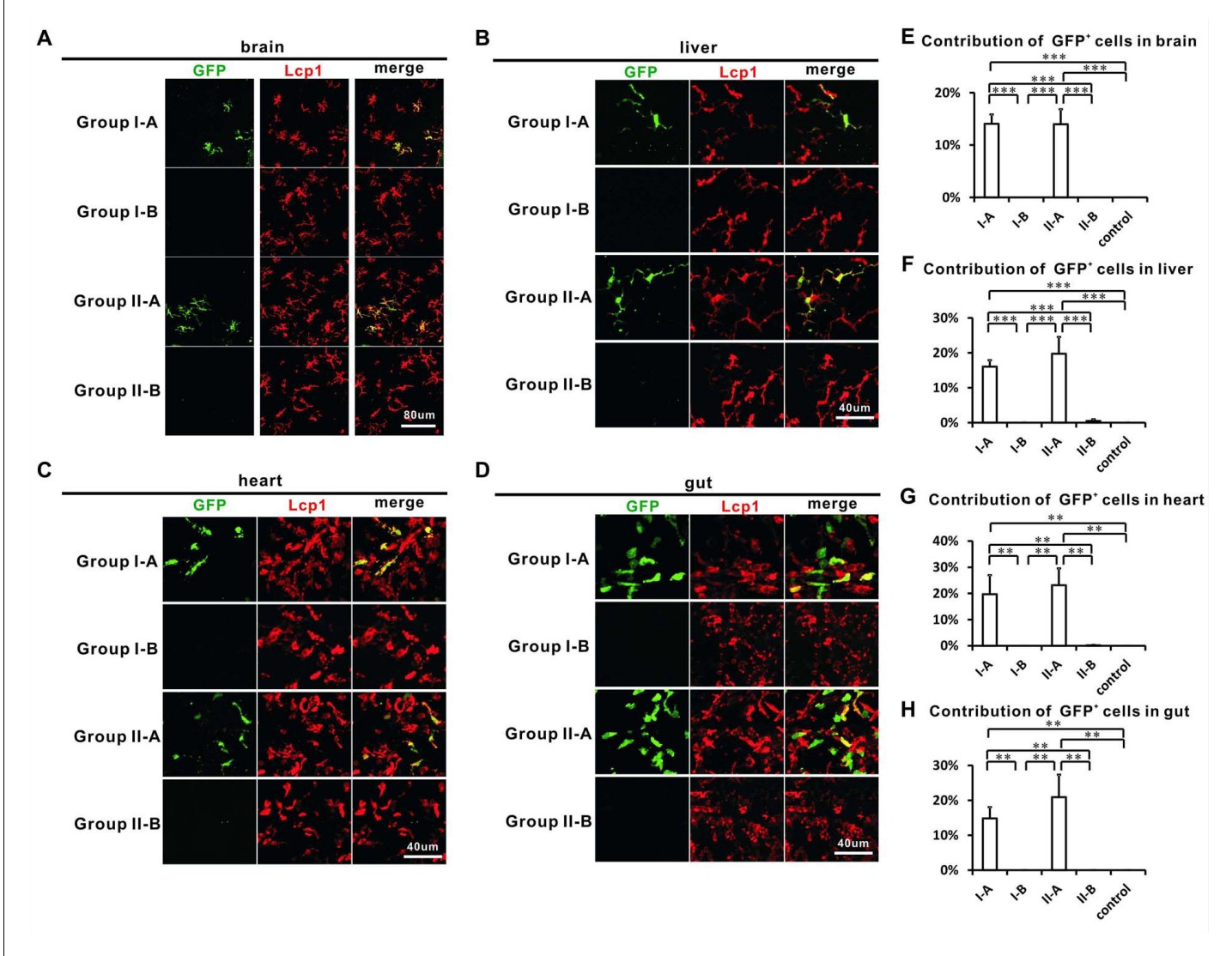

**Figure 5.** Adult tissue-resident macrophages in the brain, liver, heart and gut correlate with HSCs. (A–D) Confocal images show that GFP[+] cells are found in the brain (A), liver (B), heart (C) and gut (D) of Group I-A and II-A, but not I-B and II-B. Most GFP[+] cells are Lcp1[+] with the exception of gut GFP[+] cells which only partially overlap with Lcp1 signals. (E–H) Quantification of the contribution of GFP+ cells in all Lcp1 positive leukocytes in the brain (E), liver (F), heart (G), and gut (H) of adult Group I-A, I-B, II-A, II-B, and non-heat-shocked control group (n = 4 for each group of brain and liver; n = 3 for each group of heart and gut). Error bars represent mean SEM. ***p<0.001; **p<0.01.

DOI: https://doi.org/10.7554/eLife.36131.014

The following source data and figure supplements are available for figure 5:

**Source data 1.** Quantification data for *Figure 5E, F, G, and H*.

DOI: https://doi.org/10.7554/eLife.36131.017

**Figure supplement 1.** Other tissue-resident macrophages also correlate with HSCs.

DOI: https://doi.org/10.7554/eLife.36131.015

**Figure supplement 1—source data 1.** Quantification data for *Figure 5—figure supplement 1D and E*.

DOI: https://doi.org/10.7554/eLife.36131.016

transplanted into 2 dpf *runx1[W84X];Tg(coro1a:eGFP)* transgenic mutant embryos that were more appropriate to accept the donor cells because of the impairment of VDA-born hematopoiesis (*Figure 6A*) (*Jin et al., 2009*; *Sood et al., 2010*). The *Tg(mpeg1:loxP-DsRedx-loxP-GFP)* and *Tg (coro1a:eGFP)* transgenic lines allowed us to distinguish donor-derived *mpeg1*-DsRedx[+] cells

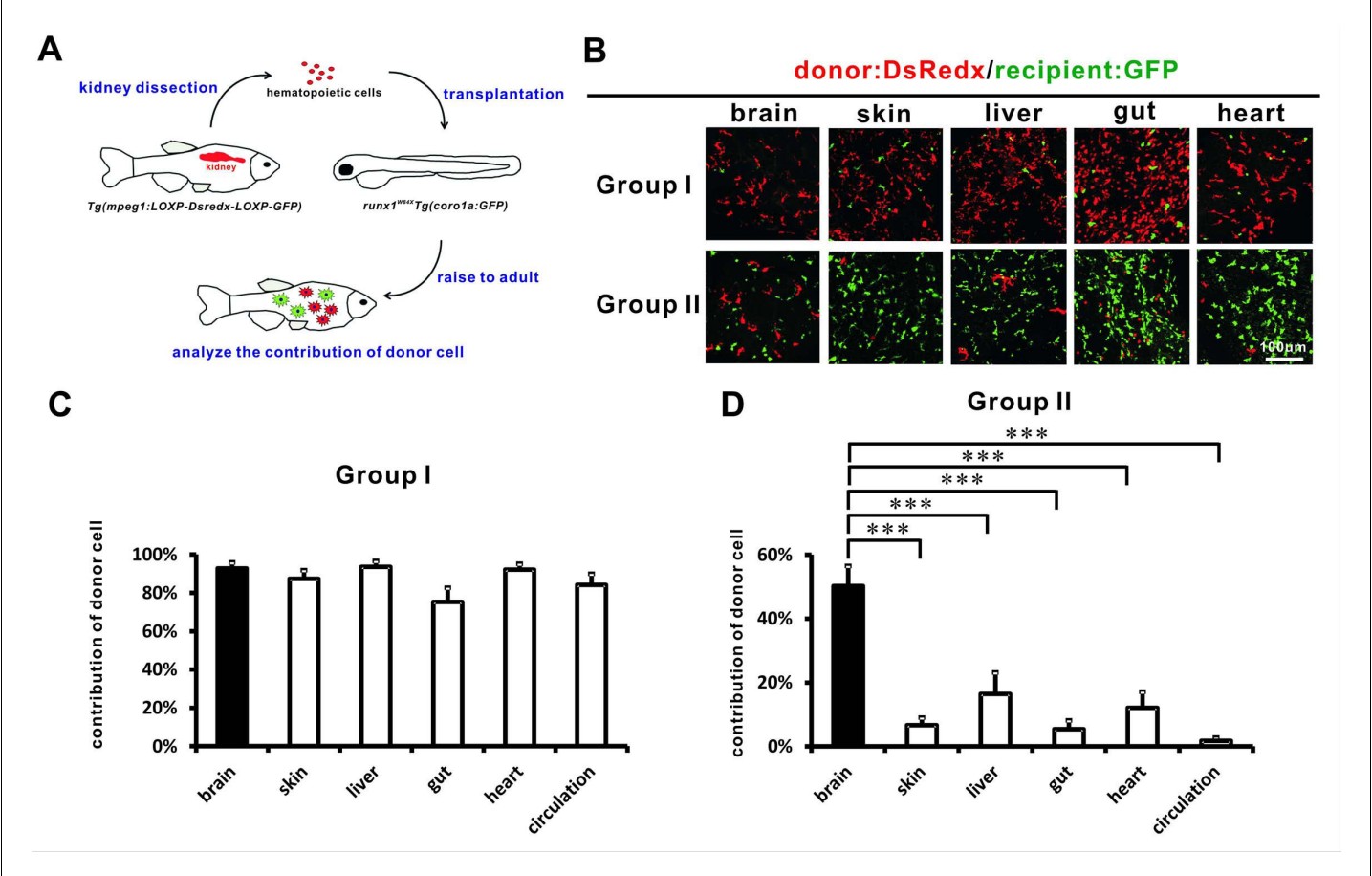

**Figure 6.** Adult tissue-resident macrophages are largely associated with HSCs during transplantation. (**A**) A schematic view of the cell transplantation experiment. Hematopoietic cells collected from the whole kidney marrow of adult *Tg(mpeg1:loxP-DsRedx-loxP-eGFP)* fish are transplanted into the circulation of 2 dpf *Tg(coro1a:eGFP);runx1^W84X* embryos. The recipients are raised to adulthood for analysis. (**B**) Donor Dsredx+ cells were predominant in the brain, skin, liver, gut, and heart of Group I fish. In Group II fish, donor cells were only significant in the brain but not in other tissues. (**C**) Quantification of the relative contribution of donor DsRedx$^+$ cells versus total fluorescent cells (Dsredx$^+$ and GFP$^+$) in brain, skin, liver, gut, heart, and circulation in Group I fish (n = 8 for brain, skin, liver, gut, and circulation; n = 7 for heart). Error bars represent mean SEM. Donor DsRedx+ cells were predominant in all tissues and in circulation of Group I fish. (**D**) Quantification of the relative contribution of donor cells (Dsredx$^+$) versus total fluorescent cells (Dsredx$^+$ and GFP$^+$) in brain, skin, liver, gut, heart, and circulation in Group II fish (n = 16 for brain, skin and circulation; n = 15 for liver and gut; n = 14 for heart). Error bars represent mean SEM. ***p<0.001. In Group II fish, although donor cells were significant in the brain, they were drastically decreased in other tissues.

DOI: https://doi.org/10.7554/eLife.36131.018

The following source data is available for figure 6:

**Source data 1.** Quantification data for *Figure 6C and D*.

DOI: https://doi.org/10.7554/eLife.36131.019

(predominantly macrophages) from host-derived *coro1a*-GFP$^+$ cells (leukocytes and their progenitors) (*Ellett et al., 2011*; *Li et al., 2012*). The transplanted recipients were then raised to adulthood (3–4 months old), and the contribution of donor-derived cells was determined by calculating the relative contribution of DsRedx$^+$ in total fluorescent (DsRedx$^+$ plus GFP$^+$) cells in the skin, liver, gut, heart, brain, and circulation. Among the 31 surviving recipients, seven did not contain detectable donor-derived DsRedx$^+$ cells (data not shown), suggesting a failure of long-term reconstitution of donor cells in these recipients. The survival of these seven recipients was likely due to the recovery of their own hematopoiesis, as shown previously (*Sood et al., 2010*). The remaining 24 recipients containing donor-derived Dsredx$^+$ cells were classified into group I and group II. Group I recipients (eight fish) were those containing abundant donor-derived DsRedx$^+$ cells in their circulation. In this group, a large number of DsRedx$^+$ donor cells were present in the brain (microglia, 93%), skin (LCs,

87%), liver (94%), gut (75%), heart (92%), and circulation (84%) (*Figure 6B and C*, *Figure 6—source data 1*), showing that these Group I recipients were well reconstituted by donor-derived hematopoietic cells, including HSCs. By contrast, Group II recipients (16 fish) contained very few DsRedx⁺ donor cells (~2%) in their circulation (*Figure 6D*, *Figure 6—source data 1*), indicating that these recipients were reconstituted by donor-derived progenitor cells but not HSCs. This group of recipients contained only small numbers of donor-derived DsRedx⁺ LCs in the skin (7%), liver (16%), gut (5%), and heart (12%). One exception was the brain, which contained ~50% of donor-derived DsRedx⁺ microglia (*Figure 6B and D*, *Figure 6—source data 1*). However, based on our previous spatial restricted cell tracing analysis, adult microglia also likely arise from the VDA-born HSCs and the HSC-independency we observed in the transplantation assay is likely due to the self-maintenance of microglia (*Ajami et al., 2007*; *Huang et al., 2018*), thus myeloid progenitors from donor kidney marrows are sufficient to generate long lasting microglia. Taken together, our data suggested that adult tissue-resident macrophages are largely associated with HSCs during transplantation.

## Discussion

In this study, by utilizing a temporospatially resolved cell fate-mapping method, we documented that LCs and other tissue-resident macrophages in the adult zebrafish were largely associated with HSCs.

Although we cannot completely rule out the possibility that LCs and other tissue-resident macrophages were derived from non-HSC progenitors, which must be linked with HSCs so closely that they cannot be distinguished by our spatial-restricted labeling, we suggested that a large portion of the adult LCs and other tissue-resident macrophages in the zebrafish were derived from an HSC origin, This conclusion differed from previous reports in mice suggesting that adult LCs and other tissue-resident macrophages arise from EMPs born in the YS (*Ginhoux and Guilliams, 2016*; *Gomez Perdiguero et al., 2015*; *Hoeffel et al., 2015*; *Kierdorf et al., 2015*; *Schulz et al., 2012*). Also, our conclusion was different from another report, in which microglia was suggested to originate from non-HSC progenitors (*Sheng et al., 2015*). One possible explanation for the different conclusions could be the species differences. Zebrafish might differ from mice in terms of the origin of tissue-resident macrophages. Another explanation was that this discrepancy was due to the different fate-mapping methods used in these studies. Unlike the high temporospatial resolved fate-mapping used in the current study, the fate mapping studies conducted in mice relied on the promoter-controlled CreER-*loxP* tracking system, in which the labeling resolution was determined by the specificity of the promoter (*Gomez Perdiguero et al., 2015*; *Hoeffel et al., 2015*; *Schulz et al., 2012*; *Sheng et al., 2015*). Thus, confounding conclusions could be obtained with the different promoters as drivers. In fact, by utilizing the c-*Kit-MER-Cre-MER-loxp* reporter system, Sheng et al., showed that most LCs and some other tissue-resident macrophages in adult mice arose from HSCs, which differed from the EMP origin obtained from previous fate-mapping studies using the *Runx1-MER-Cre-MER-loxp* and *Tier2-MER-Cre-MER-loxp* reporter systems (*Gomez Perdiguero et al., 2015*; *Hoeffel et al., 2015*; *Schulz et al., 2012*; *Sheng et al., 2015*). Interestingly, evidence supporting an HSC origin of LCs in mice can be found in reports supporting an EMP origin of LCs. For instance, a substantial number (30%) of LCs were recorded in 1-year-old *Flt3*^Cre^*Rosa26*^YFP^ reporter mice (*Gomez Perdiguero et al., 2015*), in which HSCs and HSCs-derived hematopoietic progenitors were labeled upon Cre-mediated excision (*Christensen and Weissman, 2001*; *Gomez Perdiguero et al., 2015*). Therefore, it was conceivable that a substantial portion of adult LCs and other tissue-resident macrophages in mice were derived from conventional HSCs born in the AGM. Further in-depth studies in mice with high temporospatially resolved fate mapping will be necessary to clarify this issue.

Our study also reinforced the hypothesis that the density of HSCs gradually decreased from the anterior to the posterior of the VDA. An important question is which molecular determinants dictate the fate decision of HSCs versus non-HSCs in the VDA. We speculated that the Wnt-Notch signaling pathways and inflammatory signaling cascades, which are involved in HSC specification in the VDA (*Clements et al., 2011*; *Espín-Palazón et al., 2014*; *He et al., 2015*; *Li et al., 2014*), were likely to have critical roles in HSC fate determination. Several previous studies in mice have shown that the aorta endothelium isolated from different time windows of mouse embryos appears to manifest, at least under in vitro culture conditions, distinct differentiation potentials, including non-HSC features (*Bertrand et al., 2005*; *North et al., 2002*; *Ody et al., 1999*; *Taoudi and Medvinsky, 2007*;

*Yoshimoto et al., 2011*). These studies suggested that, perhaps similar to zebrafish, this non-HSC population of cells also exists in the mammalian aorta endothelium. All these issues warrant further studies, and the zebrafish system provides an excellent platform to address these questions.

# Materials and methods

## Key resources table

| Reagent type (species) or resource | Designation | Source or reference | Identifiers | Additional information |
|---|---|---|---|---|
| strain, strain background (*Danio rerio*) | *Tg(mpeg1:loxP-DsRedx-loxP-GFP)* | doi: 10.1016/j.devcel.2016.06.018. | | |
| strain, strain background (*Danio rerio*) | *Tg(coro1a:loxP-DsRedx-loxP-GFP)* | doi: 10.1016/j.devcel.2015.08.018. | | |
| strain, strain background (*Danio rerio*) | *runx1*$^{W84X}$ mutant | doi: 10.1242/dev.029637. | | |
| strain, strain background (*Danio rerio*) | *cmyb*$^{hkz3}$ mutant | doi: 10.1182/blood-2011-03-342501. | | |
| antibody | Anti-GFP | Abcam | ab6658 | 1:400 Overnight 4°C |
| antibody | Anti-DsRedx | Clontech | 632496 | 1:100 Overnight 4°C |
| antibody | anti-Lcp1 | doi: 10.1242/dev.029637. | | 1:400 Overnight 4°C |

## Zebrafish

Zebrafish were maintained according to standard protocol (*Westerfield, 1995*). AB wild-type, *cmyb*$^{hkz3}$ mutant (*Zhang et al., 2011*), *runx1*$^{W84X}$ mutant (*Jin et al., 2009*; *Sood et al., 2010*), *Tg (coro1a: loxP-DsRedx-loxP-GFP)* (*Xu et al., 2015*), *Tg(hsp70:mCherry-T2a-CreER*$^{T2})^{#12}$(*Hans et al., 2011*), *Tg(krtt1c19e:EGFP)* (*Lee et al., 2014*), and *Tg(mpeg1:loxP-DsRedx-loxP-GFP)* lines were used in this study.

## Cell labelling with the IR-LEGO-CreER-loxP system

The experiment was performed according to the previous report (*Xu et al., 2015*). In brief, the RBI, VDA or PBI region of fish embryos was heat-shocked with IR laser at 65–80 mw for two minutes per hit. The heat-shocked embryos were then treated with 4-OHT (10 µM, Sigma) for 17–22 hr. After washing with egg water, the 4-OHT-treated embryos were raised and analyzed at the desired stage.

## Generation of transgenic lines

The 4.3 kb *mpeg1* promoter, *loxP* sequence, coding regions of DsRedx or GFP were cloned into the pTol2 vector to generate the *mpeg1:loxP-DsRedx-loxP-GFP* construct. The resulting construct was injected, together with the mRNA of transposase, into 1 cell stage fertilized embryos (*Kawakami et al., 2000*). The embryos were raised to adult and the transgenic founder was identified by direct observation under a fluorescent microscope.

## Fish dissection, antibody staining, and imaging

The experiment was performed as previously described (*Xu et al., 2015*). In brief, fishes were fixed in 4% PFA at 4°C for 1–2 days. After washing, the fish were either directly subjected to whole-mount antibody staining (for 4 dpf fish) or dissection (for adult fish). The adult skin and other tissues were applied to whole-mount antibody staining as previously described (*Barresi et al., 2000*; *Jin et al., 2006*). The primary antibodies included anti-GFP antibody (ab6658, Abcam), anti-DsRedx antibody (632496, Clontech), and anti-Lcp1 antibody (*Jin et al., 2009*). The secondary antibodies were Alexa 488-anti-goat antibody (A11055 Invitrogen) and Alexa 555-anti-rabbit antibody (A31572 Invitrogen). Images were taken under Leica SP8, Zeiss LSM710, or Zeiss LSM880 confocal microscope. The quantification was performed manually.

## Transplantation of whole kidney marrow (KM) cells

The transplantation experiment was performed as described (*Traver et al., 2003*). In brief, the kidneys of adult *Tg(mpeg1:loxP-DsRedx-loxP-GFP)* fish were dissected and hematopoietic cells were dissociated by vigorous aspiration. After washing with PBS containing 2% fetal bovine serum (FBS), the KM cells were suspended in PBS containing 2% FBS, DNaseI and heparin. About several hundred cells were transplanted into the circulation of each 2 dpf *runx1*$^{W84X}$;*Tg(coro1a:eGFP)* embryos. The

recipients were raised to adult and the relative contribution of donor cells were determined by DsRedx$^+$ versus total fluorescent cells.

## Flow cytometry

Flow cytometry was performed as described (*Traver et al., 2003*). In brief, cells in the circulation were collected in PBS with 2% FBS from the gill of adult fish by aspiration with a pipette. The collected cells were analyzed in BD FACSAria IIIu flow cytometer or Beckman Coulter CytoFLEX.

## Statistical analysis

The F-test of two-sample for variances and t-test of two-sample assuming equal/unequal variances were tested by the Data Analysis tool in the Excel software (Microsoft Corporation). Two-tailed p-values are used for all t-tests. One-Way ANOVA and Post Hoc multiple comparisons: LSD (least significant difference) were performed by IBM SPSS version 20.0.

## Acknowledgements

We thank Dr. Michael Brand for sharing the *Tg(hsp70:mCherry-T2a-CreER$^{T2}$)* transgenic line; Dr. Thomas J. Carney for sharing the *Tg(krtt1c19e:EGFP)* transgenic line; Dr. Koichi Kawakami for providing the pTol2 vector. This work was supported by the National Natural Science Foundation of China (31229003), by the General Research Fund from the Research Grants Council of the HKSAR (16102414; 16103515; HKUST5/CRF/12R; AoE/M-09/12; and T13-607/12R) and by the Innovation and Technology Commission of the HKSAR (ITCPD/17–9).

## Additional information

### Funding

| Funder | Grant reference number | Author |
|---|---|---|
| National Natural Science Foundation of China | 31229003 | Zilong Wen |
| Research Grants Council, University Grants Committee | 16102414 | Zilong Wen |
| Innovation and Technology Commission | ITCPD/17-9 | Zilong Wen |
| Research Grants Council, University Grants Committee | 16103515 | Zilong Wen |
| Research Grants Council, University Grants Committee | HKUST5/CRF/12R | Zilong Wen |
| Research Grants Council, University Grants Committee | AoE/M-09/12 | Zilong Wen |
| Research Grants Council, University Grants Committee | T13-607/12R | Zilong Wen |

The funders had no role in study design, data collection and interpretation, or the decision to submit the work for publication.

### Author contributions

Sicong He, Jiahao Chen, Conceptualization, Data curation, Formal analysis, Validation, Investigation, Methodology; Yunyun Jiang, Data curation, Formal analysis, Validation, Investigation; Yi Wu, Formal analysis, Validation, Investigation; Lu Zhu, Wan Jin, Changlong Zhao, Formal analysis, Investigation; Tao Yu, Tienan Wang, Investigation; Shuting Wu, Resources; Xi Lin, Data curation, Methodology; Jianan Y Qu, Supervision; Zilong Wen, Conceptualization, Supervision, Funding acquisition, Writing—original draft; Wenqing Zhang, Conceptualization, Resources, Supervision, Investigation, Writing—original draft, Project administration; Jin Xu, Conceptualization, Data curation, Formal analysis, Supervision, Validation, Investigation, Methodology, Writing—original draft, Writing—review and editing

**Author ORCIDs**

Sicong He (iD) http://orcid.org/0000-0002-0399-3904

Jin Xu (iD) http://orcid.org/0000-0002-6840-1359

**Decision letter and Author response**

Decision letter https://doi.org/10.7554/eLife.36131.022

Author response https://doi.org/10.7554/eLife.36131.023

## Additional files

**Supplementary files**

• Transparent reporting form

DOI: https://doi.org/10.7554/eLife.36131.020

**Data availability**

All data generated or analysed during this study are included in the manuscript and supporting files. Source data files have been provided for all figures and supplementary figures.

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
