## [Decision Letter]

Thank you for submitting your article "Adult zebrafish Langerhans cells arise from hematopoietic stem cells" for consideration by *eLife*. Your article has been reviewed by three peer reviewers, including Leonard I Zon as the Reviewing Editor and Reviewer #1, and the evaluation has been overseen by Sean Morrison as the Senior Editor.

The reviewers have discussed the reviews with one another and the Reviewing Editor has drafted this decision to help you prepare a revised submission.

Summary:

This a very interesting paper on the origin of Langerhan's cells using the zebrafish as a model. The authors have used fate mapping technology, and have seen that there are distinct sources of Langerhans cells based on developmental timing. This helps provide a broad understanding of these cells.

Essential revisions:

The authors should pay particular attention to the questions raised by reviewers #1 and #2 about the origin of the cell from stem cells. The authors should consider marrow transplants to test whether adult HSCs can give rise to these cells. The second reviewer offers an opportunity to do a time course to help with the VDA as a potential source of the cells. These approaches will help make more definitive statements about the stem cell origin. Otherwise, the authors may have to back off of some of the claims about stem cells. Rephrasing of these points, and mentioning the caveats, was suggested by all the reviewers.

*Reviewer #1:*

The article entitled "Adult zebrafish Langerhans cells arise from hematopoietic stem cells" by He et al., identify distinct origins for embryonic and adult Langerhans cell (LC) populations. Using laser-guided Cre-Lox lineage tracing to induce GFP labeling in mpeg1+ populations from distinct hematopoietic sites, the authors show that at 4 days post fertilization (dpf), morphologically identifiable LCs are derived from cells labeled in the rostral blood island (labeled at 14-16 hours), whereas at 4 months post fertilization, LCs are mostly derived from cells labeled in the ventral wall of the dorsal aorta (VDA, labeled at 28 hours). The authors then go on to show that in fish mutant for Runx1 or cMyb, genes important for HSPC development, there is a concomitant loss of the numbers of GFP^+^ LCs. Through random spot laser-labeling along the VDA, the authors show that there is a correlation of higher rates of LC labeling with instances of putative HSC labeling – as determined by retained GFP^+^ label in circulating blood cells in adult fish. Moreover, the authors show that the anterior region of the VDA appears to more frequently give rise to LCs, which agrees with their previous work suggesting that this region may preferentially give rise to HSCs. Next, the authors repeated their spot labeling in the VDA using a pan-leukocyte lineage trace and follow labeled cells to the thymus at 5 dpf. From here, the authors split animals into two groups (label in thymus vs not) and grow them to adulthood and assessed GFP^+^ contribution to peripheral blood. Those with peripheral blood labeling were considered "HSC labeled", and these animals had significantly higher numbers of GFP^+^ LCs than non-HSC labeled, regardless of if there was GFP signal detected in the thymus at 5dpf or not. The authors interpret this correlation with HSC labeling to mean that adult LCs are derived from HSCs, while labeling of thymus-colonizing progenitor cells does not. Finally, the authors repeat their initial labeling of distinct hematopoietic regions of the embryo and identify GFP^+^ macrophages in liver, heart, gut, and brain only after labeling was induced in the VDA. This study contributes to the ongoing debate on origins of tissue resident macrophage populations in development. The manuscript is logically organized, and if the authors address the comments below, then the work would be suitable for publication in *eLife*.

1) The authors do not directly show that HSCs give rise to all adult LCs. Rather, they show a correlation between HSC labeling and higher rates of LC labeling. To strengthen the relationship between HSCs and LCs the authors should pursue experiments to show multilineage contribution and LC development from single cells. Adding a marrow transplant would be useful. This would resolve the issue of potentially labeling LC-progenitors and assuming an HSC origin in their existing work. The authors should comment on non-GFP^+^ LCs in all labeling experiments – notably in Figure 1D – what are the DsRedx^+^ GFP^-^ LCs? Given that the entire VDA is labeled, these must either be LCs derived from a non-VDA source, or this may reflect the limits to laser-induced Cre-Lox labeling. If so, the authors should comment on the relative efficiency of their labeling system. Without directly showing contributions from HSCs specifically, the authors should change the title of their paper that is less definitive in an HSC-origin for LCs – for example "Adult zebrafish Langerhans Cells arise from cells in the ventral dorsal aorta."

2) The use of Runx1 and cMyb mutants would strengthen the argument for HSC origin of LCs but only if the effect on LC numbers can be attributed to changes in HSCs. Runx1 and cMyb are expressed in hematopoietic progenitors as well as stem cells. Moreover, these genes are more widely expressed than just the hematopoietic system – any effect identified in these mutants may not necessarily be due to HSPC dysfunction. The authors should use another method to quantify changes in LC number in these mutants (rather than only cells/mm^2^) as reduced numbers of LCs derived from the VDA in these mutants may not necessarily be due to reduced numbers of HSPCs and could, for example, be due to migratory defect. To show that the LC phenotype in the Runx1 and cMyb mutants is because of defects in HSPCs specifically, the authors should perform rescue experiments to restore wildtype Runx1 and cMyb expression in HSPCs.

*Reviewer #2:*

The study by He and colleagues, entitled "Adult zebrafish Langerhans cells arise from hematopoietic stem cells", used lineage tracing assays in zebrafish to investigate the developmental origins of Langerhans cells in the skin. The authors use a heat shock promoter to induce expression of a reporter construct in a spatio-temporally restricted manner, which they propose has less bias than prior studies using lineage-associated promoter-driven constructs. Using this method the authors find that the majority of labeled cells in the adult appear to derive from populations within the dorsal aorta, rather than the earlier sites of hematopoiesis. Through correlational analyses and use of HSC knockout lines, the authors propose that the origin of the Langerhans cells, as well as other tissue resident macrophage populations present in the adult, are the embryonic HSCs. This data seems in conflict with many prior studies in the mouse. While the analysis is compelling, some caveats exist that should be addressed prior to publication.

1) One major concern regarding the study is the time windows that the authors use for labeling their populations of interest. While the labeling of the rostral blood island (RBI) and ventral dorsal aorta (VDA) occur during the window when stem and/or progenitor cells are known to actively be produced, the posterior blood island is labeled much earlier than the authors state that EMPs are documented to arise. While the authors indicate that this was done prior to the onset of the heart beat to avoid potential contamination from circulating cells derived from other locations, it unfortunately makes it difficult to know if they actually labeled anything that even had the potential to make the Langerhans population (precursors to the progenitors not yet specified or potentially not even present in the tissue yet). The fact that even the larval population of skin macrophages wasn't labeled, makes this quite suspect. At this point is the primary distinction they are hoping to make with this paper (EMP vs. HSC origin), it would be very important to know what the authors would find if the labeling occurred at a later timepoint, closer to the window of EMP production as performed in the other hematopoietic sites.

2) While certainly not the authors fault, a related concern is the fact that a full characterization of the sites capable of making an EMP-like cell is missing in zebrafish. While these cells are certainly produced in the PBI, to date there have not been careful analyses indicating that cells with similar erythroid/myeloid bias cannot arise in the VDA. Recent work in the mouse indicates a number of differentiation-restricted progenitors can arise in the dorsal aorta. Given that the PBI, VDA and later even the CHT are all part of the same vascular bed in zebrafish, and not as spatially restricted as the YS vs. embryo in the mouse, it is difficult to say with certainty in the absence of robust functional studies that no EMPs are present. The authors attempt to partially address this by labeling different sections of the VDA, but given that ¾ of the region appeared to have HSC-potential as indicated by the presence of labeled circulating cells in the adult, it is hard to eliminate this concern. It may be better to rephrase the title and text to emphasize that the Langerhans cells arise from the ventral dorsal aorta without overstating the proof of HSC origin.

*Reviewer #3:*

Overall, this is a straightforward and convincing study linking the origins of Langerhans Cells with the first HSCs born in the embryo. The methods used in this paper are superior to those used in murine studies, as they are more direct since they feature spatial marking in addition to labeling by lineage-specific Cre recombination. I have no major concerns with the data presented and conclusions reached.

---

## [Author Response]

Essential revisions:The authors should pay particular attention to the questions raised by reviewers #1 and #2 about the origin of the cell from stem cells. The authors should consider marrow transplants to test whether adult HSCs can give rise to these cells. The second reviewer offers an opportunity to do a time course to help with the VDA as a potential source of the cells. These approaches will help make more definitive statements about the stem cell origin. Otherwise, the authors may have to back off of some of the claims about stem cells. Rephrasing of these points, and mentioning the caveats, was suggested by all the reviewers.

In the revised manuscript, we have included the marrow transplant analysis, which we performed previously, showing that the transplanted adult kidney marrow cells are indeed capable of giving rise to adult LCs in the recipients (subsection “Adult tissue-resident macrophages are largely associated with HSCs during transplantation”). In addition, we have also included cell-tracing experiment showing that the PBI-derived myeloid cells contribute mainly to Langerhans cells at the larva stage (subsection “Adult LCs in zebrafish arise predominantly from the VDA region”). Collectively, these data show that myeloid cells born in the PBI predominantly contribute to Langerhans cells in the larva stage, whereas the VDA-derived hematopoietic stem/progenitor cells give rise to adult Langerhans cells. To make our conclusion more precisely, we have changed our title to “Adult zebrafish Langerhans cells arise from hematopoietic stem/progenitor cells”.

Reviewer #1:[…] 1) The authors do not directly show that HSCs give rise to all adult LCs. Rather, they show a correlation between HSC labeling and higher rates of LC labeling. To strengthen the relationship between HSCs and LCs the authors should pursue experiments to show multilineage contribution and LC development from single cells. Adding a marrow transplant would be useful. This would resolve the issue of potentially labeling LC-progenitors and assuming an HSC origin in their existing work. The authors should comment on non-GFP^+^ LCs in all labeling experiments – notably in Figure 1D – what are the DsRedx^+^ GFP^-^ LCs? Given that the entire VDA is labeled, these must either be LCs derived from a non-VDA source, or this may reflect the limits to laser-induced Cre-Lox labeling. If so, the authors should comment on the relative efficiency of their labeling system. Without directly showing contributions from HSCs specifically, the authors should change the title of their paper that is less definitive in an HSC-origin for LCs – for example "Adult zebrafish Langerhans Cells arise from cells in the ventral dorsal aorta."

We fully agree that a single-cell tracing experiment could provide a definitive answer to the origin of adult LCs. Unfortunately, we cannot reach the single-cell resolution with our current setting. We have previously performed cell transplant assay, in which the adult kidney marrow cells of *Tg(mpeg1:loxP-DsRedx-loxP-GFP)* fish were transplanted into 2 dpf *runx1^W84X^;Tg(coro1a:eGFP)* transgenic mutants which are more acceptable for transplantation because of the impairment of VDA-born hematopoiesis, to examine whether adult HSCs could give rise to adult tissue-resident macrophages. Results showed that the appearance of adult tissue-resident macrophages including LCs in the recipients are highly associated with successful transplantation of HSCs. The only exception is the adult microglia, which appear to be partially independent of HSCs. However, based on our spatial restricted cell tracing analysis, adult microglia also likely arise from the VDA-born HSCs and the HSC-independency we observed in the transplantation assay is likely due to the self-maintenance of microglia, thus myeloid progenitors from donor kidney marrows are sufficient to generate long lasting microglia. We have included the cell transplantation experiment in our revised manuscript (subsection “Adult tissue-resident macrophages are largely associated with HSCs during transplantation”).

Regarding the efficiency of the infrared-mediated labeling, we have estimated that it is about 80%. The non-GFP^+^ LCs are likely derived from unlabeled HSPCs born in the VDA. We have included the estimated labeling efficiency and comments on non-GFP^+^ LCs in the first part of the Results (subsection “Adult LCs in zebrafish arise predominantly from the VDA region”).

Finally, as suggested, we have changed our title to “Adult zebrafish Langerhans cells arise from hematopoietic stem/progenitor cells”.

2) The use of Runx1 and cMyb mutants would strengthen the argument for HSC origin of LCs but only if the effect on LC numbers can be attributed to changes in HSCs. Runx1 and cMyb are expressed in hematopoietic progenitors as well as stem cells. Moreover, these genes are more widely expressed than just the hematopoietic system – any effect identified in these mutants may not necessarily be due to HSPC dysfunction. The authors should use another method to quantify changes in LC number in these mutants (rather than only cells/mm^2^) as reduced numbers of LCs derived from the VDA in these mutants may not necessarily be due to reduced numbers of HSPCs and could, for example, be due to migratory defect. To show that the LC phenotype in the Runx1 and cMyb mutants is because of defects in HSPCs specifically, the authors should perform rescue experiments to restore wildtype Runx1 and cMyb expression in HSPCs.

We fully agree that Runx1 and cMyb are also involved in the development of lineage-specific hematopoietic progenitors as well as non-hematopoietic lineages, which may possibly affect the formation of LCs. To prove the reduction of LCs in Runx1 and cMyb mutants is indeed due to the defect of HSCs, it will require expression of Runx1 and cMyb specifically in HSCs and ask whether that could rescue the LC phenotype in Runx1 and cMyb mutants. However, this will require a HSC-specific promoter, which is currently not available in zebrafish. Instead, we performed cell transplantation assay and showed that the transplanted adult kidney marrow cells can give rise to LCs in Runx1 mutants. This observation, together with our lineage tracing analysis, which show that LCs are highly correlated with the emergence of HSPCs in the VDA, we conclude that the reduction of LCs in Runx1 mutants is likely due to the failure of HSPC development.

Reviewer #2:[…] 1) One major concern regarding the study is the time windows that the authors use for labeling their populations of interest. While the labeling of the rostral blood island (RBI) and ventral dorsal aorta (VDA) occur during the window when stem and/or progenitor cells are known to actively be produced, the posterior blood island is labeled much earlier than the authors state that EMPs are documented to arise. While the authors indicate that this was done prior to the onset of the heart beat to avoid potential contamination from circulating cells derived from other locations, it unfortunately makes it difficult to know if they actually labeled anything that even had the potential to make the Langerhans population (precursors to the progenitors not yet specified or potentially not even present in the tissue yet). The fact that even the larval population of skin macrophages wasn't labeled, makes this quite suspect. At this point is the primary distinction they are hoping to make with this paper (EMP vs. HSC origin), it would be very important to know what the authors would find if the labeling occurred at a later timepoint, closer to the window of EMP production as performed in the other hematopoietic sites.

We fully agree with reviewer #2’s opinion. The main concern is that we did not provide a positive control to show that EMPs were labelled by our PBI labeling. Actually, we previously did a time course tracing to examine the contribution to Larva LCs from RBI, VDA and PBI origins. In this experiment we found that LCs derived from PBI were detected at 6 dpf. These LCs from PBI are presumably derived from EMPs. These PBI derived LCs account for ~15% of total LCs from 8 dpf-12 dpf. In contrast, VDA derived LCs increase from ~4% at 4 dpf to over 50% at 12 dpf. This experiment suggests that we can successfully label the LC progenitors, presumably EMPs, in PBI. We have included this experiment in our revised manuscript (subsection “Adult LCs in zebrafish arise predominantly from the VDA region”).

2) While certainly not the authors fault, a related concern is the fact that a full characterization of the sites capable of making an EMP-like cell is missing in zebrafish. While these cells are certainly produced in the PBI, to date there have not been careful analyses indicating that cells with similar erythroid/myeloid bias cannot arise in the VDA. Recent work in the mouse indicates a number of differentiation-restricted progenitors can arise in the dorsal aorta. Given that the PBI, VDA and later even the CHT are all part of the same vascular bed in zebrafish, and not as spatially restricted as the YS vs. embryo in the mouse, it is difficult to say with certainty in the absence of robust functional studies that no EMPs are present. The authors attempt to partially address this by labeling different sections of the VDA, but given that 3/4 of the region appeared to have HSC-potential as indicated by the presence of labeled circulating cells in the adult, it is hard to eliminate this concern. It may be better to rephrase the title and text to emphasize that the Langerhans cells arise from the ventral dorsal aorta without overstating the proof of HSC origin.

We fully agree with reviewer #2 that we cannot completely exclude the possibility that other non-HSC progenitors could generate adult LCs. We have changed the title to “Adult zebrafish Langerhans cells arise from hematopoietic stem/progenitor cells”.